# Wire Arc Additive Manufacturing (WAAM) for Aluminum-Lithium Alloys: A Review

**DOI:** 10.3390/ma16041375

**Published:** 2023-02-06

**Authors:** Paula Rodríguez-González, Elisa María Ruiz-Navas, Elena Gordo

**Affiliations:** Departamento de Ciencia e Ingeniería de Materiales e Ingeniería Química, IAAB, Universidad Carlos lll de Madrid, Avda. De la Universidad 30, 38911 Leganés, Spain

**Keywords:** WAAM, Al-Li alloys, wire production, DED (directed energy deposition)

## Abstract

Out of all the metal additive manufacturing (AM) techniques, the directed energy deposition (DED) technique, and particularly the wire-based one, are of great interest due to their rapid production. In addition, they are recognized as being the fastest technique capable of producing fully functional structural parts, near-net-shape products with complex geometry and almost unlimited size. There are several wire-based systems, such as plasma arc welding and laser melting deposition, depending on the heat source. The main drawback is the lack of commercially available wire; for instance, the absence of high-strength aluminum alloy wires. Therefore, this review covers conventional and innovative processes of wire production and includes a summary of the Al-Cu-Li alloys with the most industrial interest in order to foment and promote the selection of the most suitable wire compositions. The role of each alloying element is key for specific wire design in WAAM; this review describes the role of each element (typically strengthening by age hardening, solid solution and grain size reduction) with special attention to lithium. At the same time, the defects in the WAAM part limit its applicability. For this reason, all the defects related to the WAAM process, together with those related to the chemical composition of the alloy, are mentioned. Finally, future developments are summarized, encompassing the most suitable techniques for Al-Cu-Li alloys, such as PMC (pulse multicontrol) and CMT (cold metal transfer).

## 1. Metal Additive Manufacturing Techniques

Additive manufacturing (AM) is defined as “a process of joining materials to make objects from 3D model data; it is usually layer upon layer, as opposed to subtractive and formative methods of manufacturing” [1]. The techniques in metal AM are divided into direct and indirect processes. The indirect processes require post-forming procedures, while the techniques included in direct AM methods use a high-power laser or electron beam as a heat source and the bonding mechanism is completely melted [2]. In addition, only two direct methods can produce metallic parts: directed energy deposition (DED) and powder bed fusion (PBF), and just one process can create an additively manufactured component from wire feedstock, direct energy deposition [3].

The aerospace industry is one of the major industries and essential players in the AM market [4] and is an impetus for innovation in aircraft materials and design structures. For instance, new aluminum-lithium alloys are being studied due to their low density, high specific modulus, and excellent fatigue, which help reduce aircraft weight and improve performance [5,6]. Therefore, the need for new production processes to obtain complex and lightweight parts combined with the application of new Al alloys has led to a growing interest in the design of new Al parts produced by AM.

Among all the AM processes, directed energy deposition (DED) and powder bed fusion are those most used in the industries for aluminum alloys. Powder-fusing systems are ideal for small and intricate parts with sophisticated and complex features, while wire-fusing systems with DED technology generally have high deposition rates but poor-quality surface finishes [7]. The advantages and disadvantages of using powder and wire in additive manufacturing are described below.

The main advantages of metallic wire are the full use of raw materials free of waste, and the easy handling and storage without any special conditions or requirements, except for a closed packing to keep the surface clean since the surface of metallic wire must be free of impurities. The main disadvantage is the lack of commercial wire with different compositions, as is the case with many high-strength aluminum alloys that are not commercially available as wire. Additionally, the chemical composition of the wire used cannot be modified, in contrast to powder feedstock, that can be mixed with alloying elements. Consequently, only a few compositions can be employed in the final part.

When powder is used in DED techniques, a high-quality one is required to avoid defects in the final part. Thus, high-quality powder is commonly used in order to achieve the nominal composition and reduce the concentration of interstitial elements, such as oxygen and nitrogen, especially those reactive with Ti and Al. However, some current studies use powder blending to tailor the final alloy composition, which provides great flexibility in alloy design [8]. Critical powder features are the packing density and flowability, which are related to the shape and size of the powder particles. The powder should spread evenly across a bed and form a gapless layer. Smooth and spherical particles flow more easily than irregular ones with a rough surface. The most commonly used methods for obtaining powders with these characteristics are gas atomization and plasma atomization.

Furthermore, the powder particles must melt completely since partial fusion of the particles could produce defects in the final product. Powder handling poses a health risk, and its storage must also be adequate, requiring specific conditions free of humidity and exposure to extreme weather. All these powder requirements are common for all bed fusion techniques.

## 2. Processes to Produce Metal Wires for AM

Production of wires is achieved by different techniques, such as casting, extrusion, and drawing. The most conventional process is casting, which uses ingots of metallic alloys. It is an elementary method that has been used for many years.

Drawing produces wires from rods, bars or plates. By means of a pulling force, the material goes through a die (a rigid tool with a wear-resistant surface), changing and reducing the cross-section. The process requires cleaning and lubrication before going through a die. Kabayama and Taguchi [9,10] defined the most important features of the drawing process as follows:Lubricant (friction coefficient, viscosity, surface treatment)Wire properties (yield stress, elastic modulus, strain rate, strain-hardening)Die geometry (reduction angle, bearing region length, reduction area, material).

The extrusion process is also an alternative for producing wires. Friction extrusion uses a heat source generated by the rotating friction between the raw material and the dies used in the process under load. When the material exhibits plastic behavior, it is forced to flow through the die.

Researchers at the University of South Carolina (USC) obtained 2050 and 2195 aluminum wires. They used friction extrusion to obtain the preliminary rod, followed by drawing, giving the wires a length range between 1.7 and 2.3 m [11]. They also showed that 15 drawing steps can be needed to obtain the final wire with a 1.6 mm diameter. The 0.1 mm step size was used to reduce 2.7 mm starting diameters to 1.6 mm. Annealing and reannealing can alleviate work hardening caused by drawing to prevent wire breaking in the posterior drawing [12]. The purpose of this work was to design wires by WAAM and explore the possibility of modifying the characteristics and initial compositions to obtain a specific final wire. This research concluded that post-extrusion drawing could improve the applicability of extruded wires in the following ways:Obtaining the desired diameterImproving surface finishIncreasing the total length

Moreover, powder consolidation in the wires is possible. The direct extrusion of powders, which is a simple metal-forming process, is being developed to obtain preliminary rods. In the subsequent step, these rods can be drawn to obtain wires. Some studies have already been carried out for this purpose, obtaining high-strength Al alloy rods by direct extrusion processes [13,14].

Another process to produce wires from powders is the Conform™ process (Figure 1). Continuous extrusion or Conform™ is used to transform powders, particulates, or waste products, such as machining swarf, into a rod/wire, by means of severe plastic deformation processes. It is possible to obtain a diameter wire of < 5 mm by means of cold drawing with 100% of the material used.

Katsas et al. [16] compared the microstructure, texture and superplastic properties developed during Conform^TM^ with a conventional extrusion for a particulate Al–4Mg–1Zr alloy. In this case, the microstructural differences of conventional extrusion from the center to the surface are observed: from a bimodal distribution of coarsely deformed and finely equiaxed grains in the center to a standard distribution of refined equiaxed grains on the surface. In the Conform^TM^, a uniform distribution of refined grains was observed throughout the cross-section. The central region is extruded without further recrystallization, whereas in the surface regions where the strain and strain rate are higher, secondary recrystallization occurs. This work demonstrated that a fully consolidated product of aluminum alloys produced from a particulate feedstock with a uniform refined grain structure could be obtained with good superplastic properties. Conform™ has been used since the 1970s for the production of copper and aluminum rods [17].

At present, there are commercial wires of some aluminum alloys sold, such as 1350, 1100, 1199, 5056, and 6061. New processes and strategies to obtain the wires are being developed to address current demands. The design and selection of the wire are essential to the performance and quality of the deposited metal to obtain a final part with the required properties. For this purpose, the properties of the wires must be studied, in which it is necessary to define the most critical ones, such as good weldability, among others. Gu et al. [18] studied the qualities of the external surface, microhardness, porosity, composition, and microstructures of 4043 aluminum wires. The starting material must be free of defects and imperfections for the application of WAAM. Alloying elements are critical for part porosity, microstructure, and final properties. Controlling small additions of alloying elements can considerably modify the welding behavior of the alloy. Apart from the development of specific wires, the selection of a suitable WAAM technique is key to building functional parts.

WAAM, whose classification is explained below, has been established as a potential technology for the large-scale production of aluminum alloy parts; however, its application is currently limited by the porosity and low mechanical properties attained.

## 3. Classification of the WAAM Techniques and Their Characteristics

The classification of the WAAM techniques based on the type of welding technique used is shown in Figure 2: gas metal arc welding (GMAW), plasma arc welding (PAW), and non-consumable tungsten electrode welding (gas tungsten arc welding, GTAW) [19].

GMAW generates an electric arc as the heat source between the consumable metal electrode and the workpiece. GMAW has two variants: metal inert gas (MIG) and metal active gas (MAG). It is ideal for producing parts on a large scale in short periods of time. These techniques basically consist of an electric arc established between the tip of a consumable wire and the part under the protection of an inert or active gas that protects the weld pool and the adjacent material. The deposition rate is 15–160 g/min [20], which is higher compared to GTAW and PAW.

GTAW and PAW generate an electric arc between the nonconsumable tungsten electrode and the workpiece. The difference with GMAW is that GTAW and PAW require a wire feed that is externally provided. The orientation and the wire feeding direction determine the characteristics and quality of the deposited material [7]. PAW and GTAW have features in common, such as a nonconsumable electrode to establish the electric arc and using an inert shielding gas without filler material [19,20]. However, PAW is a higher energy density process, where excellent stability is achieved due to the arc passing through an orifice between the cathode and anode. Consequently, the weld bead obtained by PAW tends to be narrow since it allows the welding speed to be controlled. The deposition rates are low, approximately 1 g/min, achieving a high-quality surface finish.

On the other hand, the deposition rate for the GTAW technique can be as high as 30 g/min, and the total wall width is thicker (4–15 mm) than the one obtained by PAW (2 mm). The beam’s penetration is higher with PAW, causing the fusion of the previously deposited layers and compromising the wall’s stability [20].

New techniques have been developed to improve the method according to the material used. A variation of the GMAW process, also known as a modified metal inert gas (MIG), is a cold metal transfer (CMT). CMT, known as the freezing process, compared to the other welding techniques [3], alternates cooling and heating based on a short circuit with high and low current and voltage. It applies noticeably less heat input than traditional GMAW and differs from the latter in its excellent control over penetration, a high wire melting efficiency, and a high deposition rate. Another variant is GMAW in tandem. A tandem uses two independent welding systems, synchrony, and two wires. The energy input and deposition rates are higher.

## 4. Overview of WAAM

In 1925, Baker [21] carried out studies using an electric arc as a heat source and metal wire as feedstock, but the earliest research into WAAM dates back to 1926, when Baker patented “the use of an electric arc as a heat source to generate 3D objects depositing molten metal in superimposed layers” [22]. Baker used a new technique to build a 3D object using welding, the oldest known attempt to use welding technology in additive manufacturing. The same year, Eschholz [23] used an electric arc to deposit metal to create a variety of ornamentation using only a single layer and identified the primary process variables, such as arc current, depth of penetration, travel speed, substrate material, bead width, and height.

In 1935, an electric arc was covered beneath a bed of granulated flux known as SAW (submerged arc welding). The process patented by Jones, Kennedy, and Rothermund [24] required a continuously fed consumable solid or tubular (metal-cored) electrode. In 1947, 20 years later, Carpenter et al. [25] patented the method for metal coating of metal pipes by electric fusion. This invention was used in the production of magnesium retorts, having a carbon steel base clad with a high chromium, high nickel steel alloy coating of one-half inch and a base metal thickness of one inch.

Numerous patents were accepted as the first steps in wire and arc additive manufacturing. In 1950, a process for additive manufacturing with wire deposition was described by Muller Albert et al. [26]. In 1971, Ujiie (Mitsubishi) fabricated a pressure vessel using SAW, electroslag, and TIG, and also employed different wires to provide functionally graded walls. In 1983, Kussmaul [27] used shape welding to manufacture high-quality, large, nuclear structural steel parts with a deposition rate of 80 kg/h and a total weight of 79 tons [28].

In 1990, Acheson [29] designed automatic welding equipment for weld build-up. In 1997, Dickers et al. emphasized the importance of the weld bead geometry and conducted numerous trials of singular weld beads [30], modifying parameters, such as voltage, wire feed rate, wire stick out, distance from the nozzle, etc. In addition, a feedback loop between the welder and the robot controller was shown to be essential for increasing process uniformity, according to Dickers’ software-focused research [31]. During the same year, Spencer et al. [32] focused their studies on the manufacture of thicker walls. However, there was incomplete penetration of the material, and tilting the torch to deposit several adjacent strands was unsuccessful. Finally, thanks to the combination of the GMAW technique of three axes with a Siemens controller, it was possible to build layers on a platform that could tilt and rotate.

In 2002, GMAW-WAAM was improved by using the CAD/CAM integrated system [33]. Song et al. (2007) designed the hybrid manufacturing, integrating gas metal arc welding (GMAW) and milling, reducing the dimensional accuracy from ±0.5 mm of pure WAAM to ±0.01 mm [34].

In 2010, a breakthrough was achieved by Almeida et al. [35]. They fabricated a 1-meter-long, fully dense structure of Ti6Al4V with a cold metal transfer (CMT) welding process with a high deposition rate of 3 Kg/h, and achieved a deposition efficiency of over 80%.

In 2015, Gaddes [30] used constant current/constant voltage (CC/CV), GMAW short circuit transfer (off-the-shelf unit), called CMT-WAAM, to control the weld bead geometry of the process.

In recent years, WAAM techniques have progressed, incorporating post-processing such as heat treatment of parts and research into corrosion behavior. In addition, the integration of rapid prototyping using CMI-WAAM allows bimetallic materials to be obtained and memory alloys to be shaped.

Recent studies have also focused on increasing the deposition rate. Project development by Stewart Williams at Cranfield University has made considerable contributions to the fields of aluminum and WAAM [36,37,38]. The researchers developed the WAAM process capable of creating the most significant part, a six-meter-long, 300-kg, double-sided spar created from aerospace-grade aluminum [39].

## 5. Overview of Al-Cu-Li Alloys

### 5.1. Introduction to Al-Li Alloys

The most widely used aluminum alloys in the aerospace industry are the precipitation-hardening Al-Cu alloys (2xxx series) and Al-Zn alloys (7xxx series) due to their high strength-to-weight ratio. Al-Cu-Li alloys are lightweight and ideal for reducing weight and achieving lighter and stronger parts [40]. However, Al-Cu-Li alloys are expensive compared to other aluminum alloys, with a cost that is three to five times higher than the others.

Research into Al-Cu-Li alloys began in the U.S. and Germany in the early 1920s. Although this research was interrupted for many years, in 1980, researchers developed the second generation of Al-Cu-Li alloys [41]. These alloys clearly have improved mechanical properties compared to the first-generation ones. However, the properties still could not meet most aircraft specifications in terms of thermal stability, corrosion resistance, isotropy, and weldability. The third generation of Al-Li alloys was then developed to overcome these issues [40], and some of them were finally used in the aerospace industry. For instance, AA2060 and AA2050 alloys present excellent properties, such as thermal stability, corrosion resistance, and high specific strength [42].

### 5.2. Al-Cu-Li Alloys and Their Applications

Despite the fact that the development of third-generation alloys has led to new applications in the aerospace industry, their high cost has not allowed them to be fully exploited and implemented in new applications.

Conventional aluminum alloys, such as 2024 and 7055, are still widely used. The most common aluminum alloys for aerospace are 2014, 2024, 5052, 6061, 7050, 7068, 7075, 2219, 6063, and 7475. However, the alloys that are being employed for additive manufacturing techniques are mainly 2524, 7055, 7150, and AA2024, which are currently employed in the Boeing 777.

Nevertheless, some Al-Li alloys, such as 8090, have been proposed to replace AA2014, AA7010, AA2024, and AA7075 in various locations of the EH101 helicopters for structural and nonstructural applications. In particular, in floor installations, brackets, stiffeners, longerons, bulkheads, tail cone skins, flying control structures, door rails, and seat tracks [43]. AA8090 has also been evaluated for cryogenic fuel tanks.

According to Al-Cu-Li alloys, Figure 3 shows the most used Al-Cu-Li alloys. The compositions can be found in the Aluminum Association, revised in 2018 [44]. These alloys can be divided into two main groups: high-copper and low-lithium, and low-copper and high-lithium.

First, 2050 and 2060 alloys present excellent properties, such as thermal stability, corrosion resistance, high strength, good weldability, good toughness, and lightweight [42]. A 2050 alloy has been recently evaluated for cryogenic tank applications in space launch vehicles based on its excellent fracture toughness and stability at cryogenic temperatures. In particular, 2050 is used for wing spars and ribs in commercial aircraft/launch vehicle structures. It is an alternative to legacy 7xxx-series alloys, such as 7050 and 7075, due to its higher short-transverse strength and stress corrosion cracking resistance, in addition to a significant density reduction [45,46].

A 2060 alloy was launched in 2011 by Alcao Inc. [43] and is a relatively new alloy. This alloy is used in the aerospace industry, principally in the fuselage panel [47], particularly in the fuselage skin and lower wing structures [47]. When it is used instead of the 2524-T3 or 2024-T351, it may save 7% and 14% of weight, respectively [48].

NASA has employed the 2195 (Al-Li-Cu-Zr) alloy for space components [45,49]. This alloy provides good weldability, ultra-high strength, in particular, excellent resistance to fracture at shallow temperatures, and reduced density. A 2195 alloy was developed in 1994 by NASA for the cryogenic sections of the super light weight external tank (SLWT), an integral component of the early space shuttle launch systems; mainly used in the form of a plate [50]. This alloy reduced density and increased strength, thanks to a new structural design that reduced 3400 kg of the 27,000-kg tank (a 12.5% weight savings). There were savings of millions of dollars due to this weight reduction and increased payload capacity for the shuttle [45,51].

The successive 2395 alloy provides a higher strength, and it is used as drop-in replacements for 7x5x-T77511 for fuselage frames, floor beams, and upper wing stringers [45].

Both 2198 and 2199 alloys are also used for fuselage skin application [45]. A 2198 alloy was developed in 2005 and chosen afterwards by Airbus for the fuselage skin in the A350 aircraft [52]. It is used in commercial aircraft as a sheet product.

A 2196 alloy is typically used to replace conventional alloys, such as AA7075, when weight savings are needed. This alloy is used for well-suited stringer, fuselage frame, and floor structure applications where a combination of high strength, high toughness, and low density is required. A 2196 alloy was originally developed for the Hubble solar panel frame. 2196 alloy content Ag, which increases the cost of the raw material. However, it is very promising since NASA uses it in extrusions for commercial aircraft [49] and it is also part of the Airbus A380 plane [53] due to its properties.

Both 2099 and 2199 alloys were developed for space and aircraft applications, such as aircraft fuselages and lower-wing applications; their precipitates, dispersoids, and elements have shown attractive properties [6,40]. A 2099 alloy was developed by the U.S. Air Force for its potential applications with laser beam welding [54].

Finally, 2097 alloy is used as a replacement for 2124. This alloy is used for the B.L. 19 longeron in the F16 fighter jet and the F16 bulkhead area due to the improved combination of density, modulus, corrosion resistance, high-temperature stability, and resistance to fatigue crack growth propagation compared to conventional alloys [45]. This alloy presents three new versions: 2197 (1993), 2297 (1997), and 2397 (2002). This last one has been used for Lockheed Martin F-16 bulkheads and other parts for military installations [53].

### 5.3. Influence of the Main Alloying Elements

The addition of lithium reduces the density of the alloy substantially while increasing its strength more than any other element added. Each increment of 1 wt.% of lithium decreases the density by 3% and increases the elastic modulus (E) by approximately 6% [51,55,56]. Approximately 14–16 at.% (4.7 wt.%) of Li can be wholly dissolved in solid Al at 600 °C [57].

In addition, the low atomic weight of Li (6.94 g/mol) gives it the highest heat capacity compared to any other metal (in terms of J/kg·K). These properties and its relatively good thermal conductivity make Li an appealing high-temperature heat transfer material [57]. It can be a disadvantage because aluminum also presents high thermal conductivity, making it challenging in some undercooling processes.

Al-Li precipitates tend to nucleate heterogeneously on grain boundaries in slowly cooled and overheated alloys. The precipitates sequence can be described as a solid solution α + δ’ + δ → α + δ where δ is the equilibrium phase (AlLi) and δ’ metastable phase (Al_3_Li). They can also easily nucleate homogeneously in the matrix, forming spherical precipitates, thanks to their low interfacial energy with the matrix (approximately 14 MJ/m^2^) and low precipitation activation energy [58].

The addition of copper improves strength and hardness and reduces corrosion resistance. During aging, the precipitates, solute-rich domains produce the strengthening effect in the alloy. These areas are fully coherent with the matrix, but the atomic spacings are different enough to distort the crystal lattice without discontinuity in the matrix. When the movement of dislocations is obstructed, an increase in strength is attained. Al-Cu systems follow this sequence during heating, where GP corresponds to Guinier–Preston zones [59]:

Super saturated solid solution → Cu clustering → G.P.1 → G.P. 2 (θ’’) → θ’ → θ

θ’’ is an intermediate precipitate, with a tetragonal structure, which maintains coherency with the Al matrix. When θ’ appears, the strengthening is reduced. Further heating causes the transformation from θ’ to θ, an equilibrium precipitate, with the composition Al_2_Cu.

When copper and lithium interact, they form part of the main hardening mechanism in aluminum alloys, specifically Al-Cu-Li alloys, where the most critical strengthening phases are: T1 (Al_2_CuLi), T2 (Al_6_CuLi_3_), and TB (Al_15_Cu_8_Li_3_) [53]. Some studies show that ternary T1 (Al_2_CuLi) is the primary strengthening phase, and binary phases, such as δ’ (Al_3_Li) and θ’ (Al_2_Cu), despite also being present, they contribute less to strengthening.

The precipitates found in Al-Cu-Li alloys depend on the Li relative content. θ’ and θ (Al_2_Cu) phases are formed with low Li contents (<0.6%), while the main strengthening phase, T1(Al_2_CuLi) precipitates with medium Li content (<1.4–1.5%). The δ’ precipitates appear when higher Li contents (>1.4–1.5%) are employed [60]. B. Cai et al. [61], among other studies, confirmed the presence of T1 (Al2CuLi), θ′ (Al2Cu) in the 2060 alloy.

The addition of magnesium increases strength due to solid solution strengthening. Additions greater than 1.6 wt.% Mg in Al-Cu alloys promote the formation of the metastable, incoherent S’ (Al_2_CuMg) phase near grain boundaries [62].

The addition of silver and Mg stimulates the nucleation of a fine and uniform dispersion of T1 phase [63]. However, it is essential to take into account that adding elements, such as Ag, makes the alloy more expensive.

The addition of manganese promotes the formation of precipitates of the Al_20_Cu_2_Mn_3_ phase, which controls grain size and texture during the thermomechanical processing [64]. This helps improve creep resistance and damage tolerance in fracture toughness and fatigue.

Titanium interacts with the Al matrix to form Al_3_Ti intermetallic, strengthening in addition to conventional precipitation hardening (such as in the θ’ (Al_2_Cu) phase). A study shows that adding 0.6% Ti to an Al-Cu-Mg-Ag alloy promotes the precipitation of finer and denser θ’ (Al_2_Cu) phase. It also promotes the formation of the intermetallic phase Al_3_Ti. In contrast, if the addition of Ti is increased to 1.1%, no improvements are observed; the results reveal that a high volume of Al_3_Ti phase is likely detrimental to strengthening [65].

Furthermore, it is widely used as a grain refiner. In particular, and regarding the WAAM techniques, Wang, L. et al. [66] studied the Al-Mg alloy for WAAM, using ER5356 as the filling wire and adding Ti powder between layers as a grain refiner. This study reports that the Al3Ti phase provides heterogeneous nucleation cores. The addition of Ti powder during WAAM can promote effective transformation from columnar to equiaxed grains at the interlayer interface. The ultimate tensile strength and elongation increase by 20.25 MPa and 3.13% in the horizontal direction, and by 25.89 MPa and 6.97% in the vertical direction. This study highlights the addition of Ti as a grain refiner as an excellent strategy to obtain isotropic and improved mechanical properties in the WAAM technique.

The addition of zirconium promotes the formation of coherent dispersoid β’ (Al_3_Zr) and θ’ and T1 phases and reduces the solubilities of lithium and magnesium in Al alloys [67]. Zr inhibits recrystallization and grain growth at elevated temperatures [68].

The small additions of Scandium act as a core for the formation of very refined grain microstructures. At the same time, its influence has led to the development of a new alloy, Scalmalloy (AlMgSc), developed by the Airbus Group Innovations [69]. The addition of Sc results in high mechanical properties, good ductility, and specific resistance.

Sales et al. [70] demonstrated that adding Sc in AA 5183 and AA 5356 alloys promoted the formation of Al_3_Sc intermetallic particles, with an increase of nearly 60 MPa in the ultimate tensile strength and yield stress.

The systems become more complicated when more than one alloying element is added, for instance, Al-Cu-Li, Al-Li-Mg, Al-Li-Zr, Al-Cu-Mg, or Al-Cu-Li-Mg-(Ag) systems. The effects of alloying elements in Al alloys are summarized as shown in Table 1. It should be noted that promoting the homogeneous grain and fine microstructures is ideal for obtaining good properties in the final piece by WAAM.

## 6. General Welding Problems for Aluminum Alloys

The most relevant concerns related to aluminum alloys applied to additive manufacturing techniques are: the oxidation of the material surface due to the formation of Al_2_O_3_, the solidification shrinkage (compared to ferrous metallic materials) due to the wide solidification temperature range, the high coefficient of thermal expansion (CTE), which leads to cracking phenomena due to high solidification stresses and shrinkage, high reflectivity, high thermal conductivity (which leads to rapid dissipation of heat from the scanned area and requires greater source heat), and the high solubility of hydrogen in liquid aluminum (which leads to pore formation) [72].

### 6.1. Porosity

One of the challenges for aluminum parts is producing porosity-free pieces through welding processes. The leading cause is hydrogen, which has a high solubility in molten aluminum but a poor solubility in solids. The gas dissolved in the molten metal weld is trapped during the solidification, forming pores in the solidified weld. The hydrogen content over 1 mL/100 g of aluminum produces excessive fusion zone porosity [56]. The dispersion-strengthened aluminum alloys produced via rapid solidification by powder metallurgy processing often exhibit a residual hydrogen content of over 1–5 mL/100 g of aluminum, showing an increased tendency to form a fusion zone porosity [56].

In addition, another cause of the formation of porosity comes from the aluminum oxide layer, which is rapidly formed on the surface due to its high Al affinity for oxygen. Its melting point is approximately 2050 °C, while aluminum melts at 660 °C [73]. When aluminum is melted, the oxide film is entrapped in the aluminum melting pool. This film has to be removed before welding in order to reduce the risk of porosity during welding.

The chemical composition is likewise critical in the formation of porosity. Elements such as magnesium have a beneficial effect. It has been demonstrated that magnesium at 6% raises the solubility in solids and there is up to a two-fold reduction in hydrogen absorption [74]. However, the rest of the alloying elements, such as copper and silicon, do not have the same effect, and other elements, such as zinc and lithium, which have high vapor pressures, evaporate at high temperatures and leave porosity in the material.

Different attempts have been made to reduce porosity. The combination of the inter-layer rolling process and WAAM has been studied for straight walls with 2319 and 5087 aluminum alloy wires. In this study, interlayer rolling was employed between each deposited layer, and different rolling loads were applied. It was observed that pores bigger than 5 μm in diameter were eliminated at a rolling load of 45 kN [75]. However, interlayer rolling, and post-deposition heat treatments were needed for lower applied loads, showing a reduction in the number and the percentage of the pore area.

### 6.2. Cracking

Cracking is a high-temperature phenomenon that usually takes place in alloys in the liquid melt pool. When this defect occurs during WAAM in aluminum alloys, it is known as solidification cracking; it is not the common defect known as hot-cracking, which occurs in arc welding and takes place in the partial melted zone [76].

The addition of alloying elements changes the freezing temperature of the pure metal, modifying the solidification range. Pores may also promote the generation of cracks during solidification.

When the material solidifies, the liquid with the lowest solidification point is retained in the interdendritic spaces. In this mushy zone, solidification, and thermal shrinkage exhibit stress on the solid network. This was observed in 1950; investigations such as Novikov [77], Sigworth [78], and Eskin et al. [79] described how a tear (hot-crack) initiates above the solidus temperature and propagates in the interdendritic liquid film. The fracture surface is usually smooth and sometimes shows solid bridges connecting both sides of the crack [80,81,82,83,84]. Indeed, some studies show that cracking occurs in the late stages of solidification. This happens when solid volume fractions are above 85–95% [82] and the solid phase is organized in a continuous network of grains. Different mechanisms can be found in the literature describing this phenomenon [84,85].

The addition of alloying elements to Al always leads to Al alloys with different solidification ranges, which means that Al alloys can be susceptible to cracking. The Varestraint test, developed in the 1960s by Savage and Lundin at Rensselaer Polytechnic Institute [86], is a simple test to isolate the metallurgical variables that cause cracking [87,88]. In the 1990s, Lin and Lippold [89] advanced further in the research and used Varestraints testing. Finally, they determined the magnitude of the crack-susceptible region through the measure of the temperature range in which cracking occurs. Samples were tested over a range of growing strains, and the maximum crack length in the fusion zone as a function of temperature was measured.

Singer and Jennings [90] reported the first graphs for the Al-Si system, which show crack length (inches) vs. percentage of alloying elements.

Later, Pulphrey et al. [91] investigated the cracking of many binary aluminum alloys, using the same theory as Singer et al. A maximum cracking is reached in each alloy, where the temperature interval is observed between solidus and liquidus in the solidification range [92].

The alloys can be characterized by plotting relative to the theoretical peak in cracking susceptibility under equilibrium and temperature, using the appropriate conditions (Figure 4).

These graphs are known as cracking tests; they determine the composition range within which the alloy presents a high risk of cracking. The test consists of applying a load to the weld transversely to measure the length of the crack. This measure reports the specific range of sensitivity to cracking. Some examples related to the main alloying elements of aluminum alloys are seen in Figure 5.

For instance, the 2040 and 7075 alloys containing a significant amount of Cu and Mg are susceptible to cracking, as reflected in their high values of total crack length, and thus poor weldability is expected.

Cracking susceptibility is expected for aluminum alloyed with Cu and Li content of approximately 2.5 wt.%, in particular in alloys such as 2090 and 2091. In Al-Cu-Li systems, a reduction of Li content below 2% and an increase of Cu content above 3% should reduce susceptibility, as in the case of 2060 and 2195 alloys.

### 6.3. Humping Phenomenon

Some studies, such as that by Adebayo et al. [94], have focused on the factors that promote the phenomenon of humping in WAAM. According to their research, humps are a consequence of the heat sink formed due to an accumulation of material, which reduces the penetration of the arc. To avoid this phenomenon, they suggest removing the undulation of the weld bead and using low travel speeds [94]. There are three factors that must be controlled. First, the high momentum of the backward fluid flow causes the initiation and growth of swelling. Second, the large variation of the capillary pressure of the liquid channel in the welding direction can promote the shrinkage of a liquid channel. Finally, the capillary instability makes the weld pool unstable and susceptible to collapse [95].

Some studies focused on the modification of variables in the process of reducing humps. Cold metal transfer (CMT) technology for Al-Mg alloy shows that a one-way continuous-arc trajectory presents better regularity in the part geometry, owing to the unique starting point imposed in this case; on the contrary, when using other ways, humps are formed, affecting the frequency of the deposits [96].

### 6.4. Lack of Fusion, Delamination, Residual Stresses, and Discontinuity in Weld Bead

The lack of fusion is caused by insufficient overlap between passes. Some causes include low energy input, an improper torch angle, an inappropriate weld position, and insufficient filler wire material, among others [97].

Delamination is produced when the underlying material is not completely melted, which causes delamination or separation between neighboring layers. This is due to low energy input. However, other causes of delamination by WAAM are residual stresses and contamination of parts or substrate surfaces [97,98].

Residual stress is the stress that remains in a material after all external loading has been removed. If the residual stress is large enough, it will have a dangerously negative impact on the component’s mechanical properties and fatigue life. The region between the substrate and the buildup wall experiences the highest stresses in WAAM.

Discontinuity and deviation in the weld bead are other defects that are visually observed on the surface of the weld bead and occur when high energy input is used. When the arc is unstable, there are deviations in deposition volume, and the position or orientation is inadequate [98].

### 6.5. Particular Welding Problems for Al-Li Alloys

Welding issues in Al-Cu-Li alloys are presented as two types: cracking and porosity. The first one is related to high susceptibility to cracking when the process causes high levels of thermal stress and solidification shrinkage. The Al-Cu-Li alloys were designed to optimize strength and compositions that promote good mechanical properties; they were not prepared for optimum conditions in weld crack resistance. The peak tearing susceptibility of high-purity Al-Li binary alloys appears at 2.6 wt.% lithium [99,100] (Figure 6). Therefore, Al-Li alloys with this content would be more likely to promote significant tearing during the welding process.

However, not only can lithium be the cause of the welding defects in these alloys, but it can also promote porosity. This is due to the activity of lithium during the welding process. Li atoms tend to interact with H atoms, increasing the hydrogen solubility of the liquid aluminum. Some of the components that can be formed are lithium oxide (LiO_2_), lithium hydroxide (LiOH), lithium carbonate (Li_2_CO_3_), and lithium nitride (LiN) [93,101].

Despite these problems, it is known that there are already some alloys with lithium contents close to the maximum cracking point that are used for welding, such as 2099-T83. Al-Li alloy is being used for the construction of lower-wing stringers by welding processes [102]. However, at present, the knowledge base related to the weldability of Al-Li alloys is still limited.

## 7. Suitable WAAM Techniques for Al-Cu-Li Alloys

There is scant use of WAAM in Al-Cu-Li alloys. The first research study carried out focused on post-WAAM treatments to improve properties, such as removing porosity and increasing strength. However, other problems, such as the surface oxide film (alumina), which has a higher melting point than Al, cannot be addressed by those post-treatments. For this purpose, the most recent studies have focused on a variant of the WAAM technique that works with alternating current (AC).

AC mode allows the current waveform (frequency and balance) to be modified. In one half of the cycle, the electrode will be negative (with the base plate positive), and in the next half, the electrode tip will be positive (with the base plate negative). The balance represents the relationship between the penetration (EN, electrode negative) and cleaning action (EP, electrode positive) in the percentage of the cycle. The AC mode is necessary for Al-Li alloys since it allows both actions to be worked on: cleaning the oxide layer and heating a weld bead [103]. During the cleaning, the electrons remove the oxide layer from the aluminum surface, and the shielding gas prevents new oxide from being formed during the welding process.

VP-GTAW (variable polarity gas tungsten arc welding) was applied successfully on an Al-Cu-Li alloy, in particular 2050 wire, and a thin straight wall was deposited [104]. The results showed that the inner layers consisted of refined, equiaxed grains, compared to the interlayers, which consisted of coarse, columnar grains. The secondary phases were θ (Al_2_Cu) and δ’ (Al_3_Li) phases, which were dispersed along the grain boundaries after post-deposited heat treatment. The mechanical properties, such as microhardness, in the heat-treated sample were 141HV, which showed an increase of 98.6% compared to that of the deposited one (71HV), and 55% compared to that of the wire (91HV).

Other variants of GMAW, such as cold metal transfer (CMT), are of particular interest for Al-Cu-Li alloys: CMT-P, CMT-ADV, and CMT-PADV. The CMT-P refers to pulsing recurrent, in which the high pulse current results in a higher heat input compared to the conventional CMT-ADV. The advanced path involves a polarity reversal in the short circuit; and the combination of both is CMT-PADV and pulse-advanced. CMT-PADV, developed by Fronius [20], greatly improves porosity due to the control of the polarity and the pulse cycles [105]. The most significant characteristics are the low thermal heat input, the high deposition rate, and the low spatter, which have been shown to eliminate gas pores due to an oxide cleaning effect.

Derekar et al. [106] showed results using CMT and pulsed-MIG techniques with ER5183 wire on a wrought plate substrate of Al–Mg–Mn alloy. Despite the fact that ER5183 does not contain lithium, the magnesium range makes it susceptible to common welding problems, similar to Al-Cu-Li alloys. Deposited material by pulsed-MIG showed a higher percentage of medium-size (0.21–0.30 mm) and large-size pores (>0.31 mm); and a higher pore volume for medium-sized pores (0.21–0.3 mm) than those obtained by CMT. However, small-size pore volume was more significant in CMT. The picked-up hydrogen was higher for pulsed-MIG than for CMT due to the higher arc energy and the more extensive melt at higher temperatures.

To evaluate the applicability of the alloy to WAAM techniques, it is necessary to carry out numerous weldability tests. No alloy should be discarded due to its chemical composition or the possibility of the evaporation of elements with low vapor pressure, such as Li or Zn.

The most studied 2xxx alloy is the 2024. However, new Al-Cu alloys, such as 2219 and 2319, have also been incorporated into recent studies, where they appear more susceptible to experiencing problems during the welding process. However, when the 2319 wire was deposited on 2219-T851 aluminum plates by CMT-PADV [107], no crack was observed. The process was successful because it had a significant oxide cleaning effect on the weld bead surface. Both variant processes (CMT-ADV and CMT-PADV) allow the thermal profile to be modified, resulting in a refined, equiaxed microstructure and the elimination of porosity. Gu et al. [107] showed a reduction in porosity, fine equiaxed grain structures, and uniformly distributed θ-Al_2_Cu phases for 2319 filler wire by the CMT-PADV process, which not only implies a reduction in porosity similar to the studies mentioned above, but that microstructure with good conditions is also possible for aluminum alloys.

Cong et al. [108,109] used Al-Cu wires with CMT techniques. CMT produces a high heat input and has excellent penetration; consequently, coarse columnar grains are formed, preventing the hydrogen from escaping. As a result, many pores with pore sizes varying from 10 to >100 µm were obtained.

They observed that a lower arc penetration by the variable method CMT-P decreased the escape distance for hydrogen and promoted a smaller number of pores. However, their research found that CMT-ADV mode efficiently eliminated the porosity, obtaining fine equiaxed grain structures, and concluded that the three key factors are the low heat input, a shallower penetration, and alternating polarities that produce a cleaning oxide effect (pore sizes > 50 µm). The CMT-PADV showed impressive results with no pores over 10 µm.

Cong et al. also reported the differences between wall and block structures. A block structure significantly reduces porosity due to the formation of refined microstructures using the CMT-P and CMT-ADV processes. In a block structure, the material available in the surroundings extracts heat, increasing the cooling rate, while the dissipation of heat by conduction is only possible through the underlying layers in the wall structure.

Zhang et al. [110] found that VP-CMT produced a high ultimate strength because the columnar grains transformed into equiaxed grains. As the process consists of a pulsed arc mode to produce an oscillation in combination with alternating changes in arc polarity, it promotes heterogeneous core points by breaking the dendrite arms. The range of the tensile strength anisotropy of the transverse and longitudinal tensile samples was found to be between 8 and 27%.

The combined effect of CMT, interlayer rolling, and heat treatment on the porosity and oxide cleaning surfaces of aluminum alloy by WAAM is an area of interest for many researchers. The interlayer rolling contributes to reducing porosity and greatly influences the grain structure. When rolling is applied, depending on the load, precipitates break into smaller sizes. After heat treatment, a uniform distribution of refined, smaller grains is attained. For the 2024 alloy, a preliminary study by Fixter et al. [111] showed that, with adequate control of porosity and subsequent heat treatment, the tensile properties could be improved through the WAAM process to levels comparable to those of the standard wrought products. By rolling each added layer, the interpass deformation was found to lead to further refinement in grain size and improved ductility.

New pulse technologies, such as PMC (pulse multicontrol) and PMC mix, are based on pulse-controlled spray arcs optimized by tight control algorithms. The company Fronius has managed to modify the power source platform (TPS) to obtain better welding results. Process stability is improved, heat input is reduced compared to MIG, and there is almost no spatter. Consistently, good penetration is guaranteed, there is less undercutting, and it is possible to weld more quickly and more cost-effectively. PMC Mix technology combines this pulse-controlled transfer to cycles controlled short-circuit, generating a colder phase and reducing the heat input even more [96,112].

Gomes et al. [96] also compared the PMC and CMT techniques for Al alloys. The porosity was smaller than 175 µm in diameter and was dispersed in both cases. Samples by PMC and PMC mix showed a pore fraction of 0.21% and 0.80%, respectively, much lower compared to those by CMT and CMT-P techniques, which showed 0.54% and 1.16%, respectively. These results are particularly interesting for Al-Cu-Li alloys. Regarding mechanical properties, nearly isotropic properties were obtained with a difference of 13 MPa higher in the longitudinal direction of the deposition. Therefore, the small pore fractions and the regularity of the deposits obtained with PMC and PMC Mix, compared with CMT and Pulsed-MIG, point to the benefits of these techniques.

Synchro-feed welding [113] is the most advanced technology and produces a high-quality and high-speed weld without requiring a post-weld cleanup. It incorporates a driven wire feeder within the torch body, advancing the welding wire forward to create an arc. It then retracts the wire while synchronizing with a specialized weld current waveform that extinguishes the arc to make consistent droplet transfer with virtually zero weld spatter. This way, it combines a speedy wire feed control and ultra-low spatters with ultra-low heat input and ultra-low smut. Finally, it results in a very neat, precise weld laid down at a rate of up to 100 inches (254 cm) per minute using a welding current of up to 300 amps [113].

Another improvement was found by Zhang et al. [114] y means of the workpiece vibrating in combination with the VP-CMT technique. The most conclusive results were the refined grain and the homogenized grain distribution with increased vibration. The average grain size decreased due to the over-threshold bending stresses induced by workpiece vibration, breaking the dendrite arms and evolving into more nuclei. In this way, the workpiece vibration was able to significantly reduce the porosity from 6.66% to 1.52% and improve the mechanical properties; the workpiece vibration induced the molten pool stirring, which removed the fine grain zone of the interlayers and the pore defects [114]. Therefore, the combination of vibration and WAAM techniques will help to significantly reduce porosity in Al-Cu-Li alloys.

## 8. Conclusions

WAAM is a promising manufacturing alternative to conventional manufacturing. The high deposition rates make it an ideal technology for manufacturing large metal components. 

However, its use is constrained by the limited availability of commercial wire. Meanwhile, although WAAM techniques are under continuous development, as previously explained for the latest advances, the material and desired wire composition remain unaddressed by researchers. WAAM technology and material should have a parallel development in order to achieve potential breakthroughs for new aerospace components.

This review summarizes the features of conventional and innovative processes of wire production, such as the extrusion and conform processes, and seeks to encourage further research into the development of new metallic wires for WAAM.

In particular, this review provides the features of the main Al-Cu-Li alloys that are of great interest, along with their specific industrial applications, aimed at fostering and promoting the selection of the most suitable compositions. The addition of different alloying elements to aluminum has been discussed to find the most suitable wires for WAAM, focusing on lithium because of its particular strengthening benefit compared to other alloying elements. Additionally, alloying elements, such as Cu and Mn strengthen the alloy by age hardening, while Ti and Zr directly influence grain size reduction.

Finally, this review addresses specific issues for aluminum, such as porosity, cracking, and humping phenomena, and the specific defects in WAAM, such as lack of fusion and delamination, with particular interest in the role that lithium plays in porosity and cracking problems. Future strategies and developments to improve the results obtained in WAAM have been presented. The possibility to design and develop new metallic wires for the most precise and sensitive WAAM technologies, such as CMT, is open and attainable through future research.

## Figures and Tables

**Figure 1 materials-16-01375-f001:**
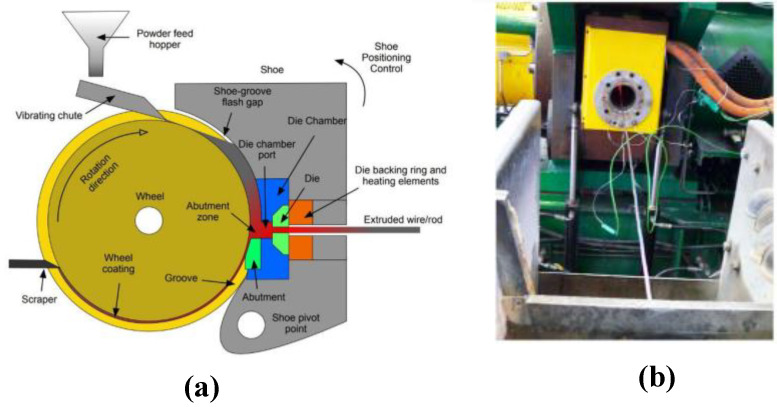
(**a**) Scheme of the Conform machine. (**b**) Extruded wire exiting from the Conform machine and entering the water trough for quenching [15].

**Figure 2 materials-16-01375-f002:**
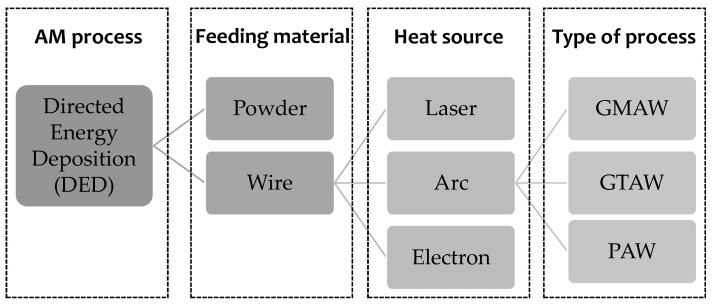
Classification of the DED process [3].

**Figure 3 materials-16-01375-f003:**
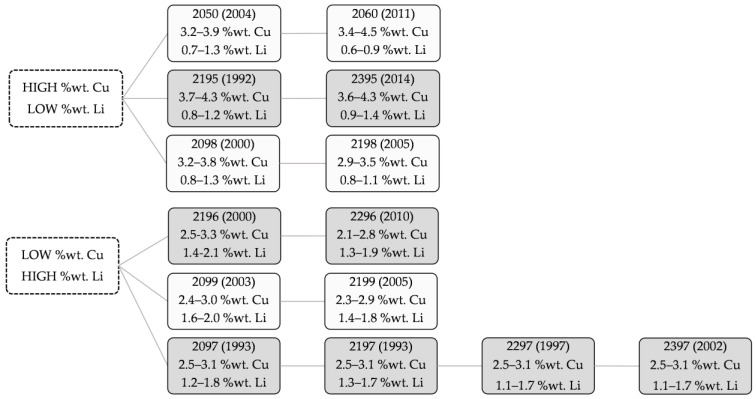
Scheme of the most used Al-Cu-Li alloys.

**Figure 4 materials-16-01375-f004:**
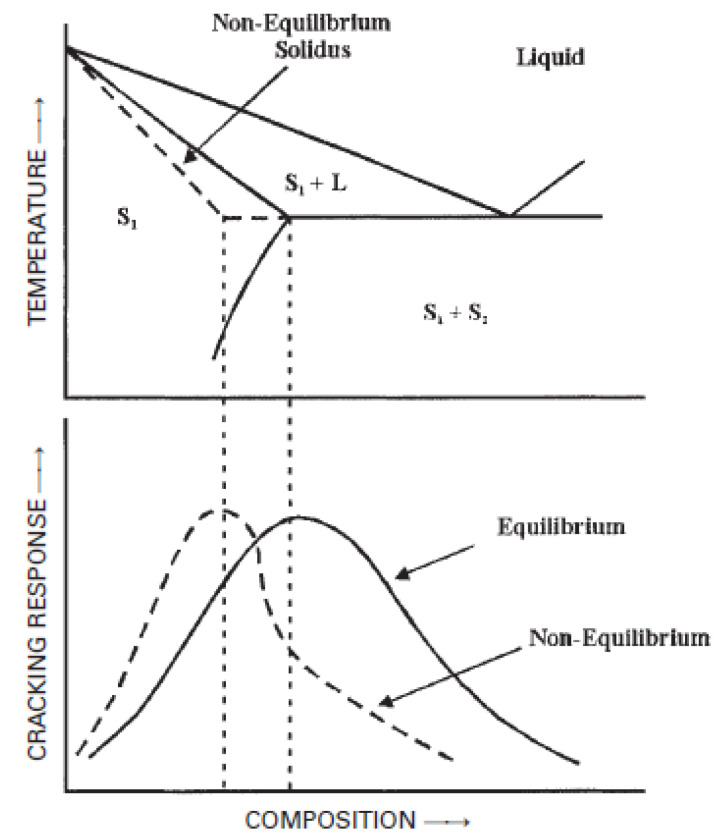
Weld metal cracking susceptibility as a function of composition in a simple binary alloy system: equilibrium and nonequilibrium solidification conditions are considered [93].

**Figure 5 materials-16-01375-f005:**
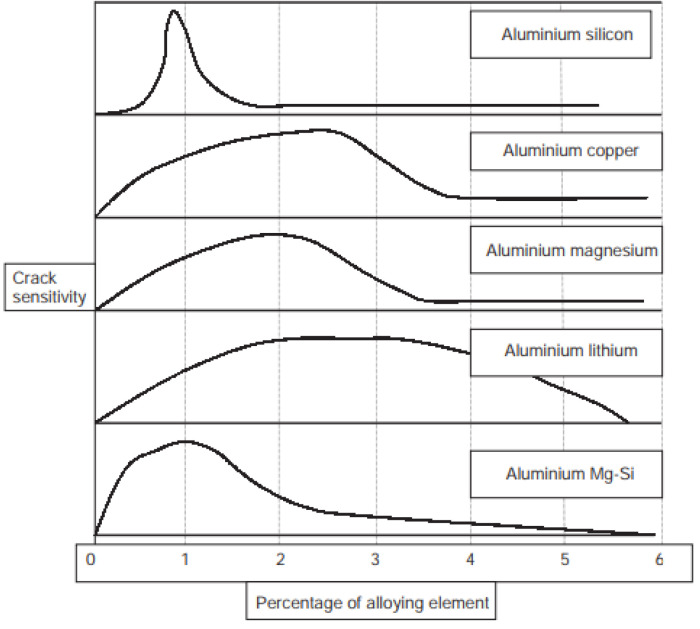
Graphs of crack sensitivity for different alloying elements in an Al matrix [74].

**Figure 6 materials-16-01375-f006:**
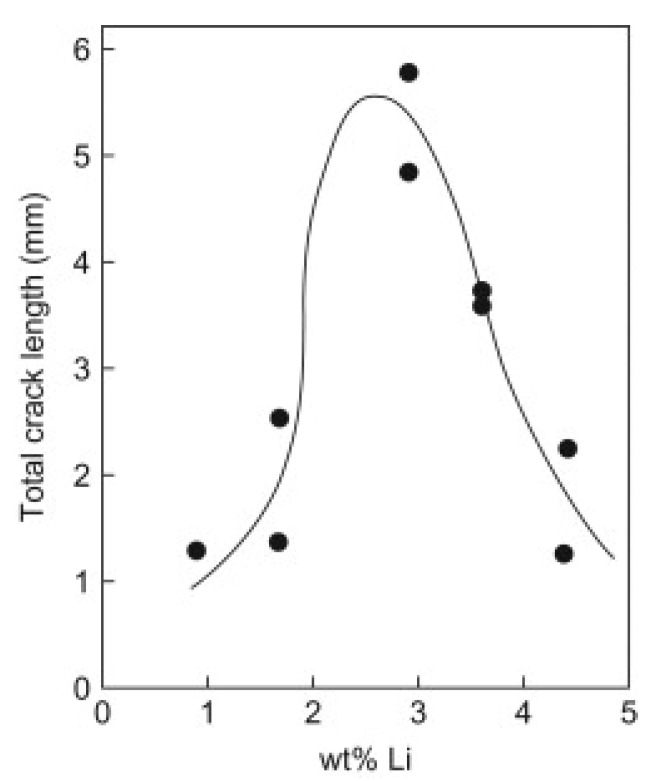
Independent peak of cracking susceptibility of Al-Li binary alloys [100].

**Table 1 materials-16-01375-t001:** Effects of alloying elements.

Elements	wt.%	Physical Properties	Mechanical Properties	Chemical Properties	Others
**Li**	0.6–2.1	↓ density ↓ weight	↑ strength ↑ hardness by age hardening		δ’ (Al_3_Li) [57] promote T1, T2, θ’
**Cu**	2.4–4.5		↑ strength ↑ hardness by age hardening	↓ corrosion resistance	θ (Al_2_Cu), T1 (Al_2_CuLi), T2 (Al_6_CuLi_3_), TB (Al_15_Cu_8_Li_3_) [71,53]
**Mg**	0.1–1.1		↑ strength ↑ hardness by solid solution	↑ corrosion resistance and weldability	Al_2_ (Cu, Li-Mg) S’ (Al_2_CuMg) [62]
**Ag**	0.05–0.6		↑ strength ↑ hardness by age hardening	↑ resistance to stress corrosion	promote T1 [63]
**Mn**	0.1–0.5		↑ strength ↑ hardness by age hardening ↑ creep resistance and damage tolerance		Al_20_Cu_2_Mn_3_ [64]
**Ti**	0.1–0.12		↑ strength by grain size reduction		Al_3_Ti [65,66]
**Zr**	0.04–0.2		↑ strength by grain size reduction	↓ resistance to stress corrosion cracking	Al_3_Zr [67] promote θ’ and T1
**Sc**			↑ strength by grain size reduction		Al_3_Sc

## Data Availability

Not applicable.

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
