# Peer review of "Wire Arc Additive Manufacturing (WAAM) for Aluminum-Lithium Alloys: A Review"

_materials, 2023, doi:10.3390/ma16041375_

Round 1

Reviewer 1 Report

The article is a review of wire-arc processes for aluminum-lithium alloy parts.
The following topics are discussed:
1. Metal additive manufacturing techniques
2. Processes to produce metal wires for AM
3. Classification of the WAAM techniques and their characteristics
4. Overview of WAAM
5. Overview of Al-Cu-Li alloys
6. General welding problems for aluminum alloys
7. Suitable WAAM techniques for Al-Cu-Li alloys

A large part of the article reviews WAAM processes for aluminum alloys, whereas the title only announces a review of aluminum lithium alloys. To make the reading of the article more fluid and to match the given title, some parts could be lightened, such as
1. Metal additive manufacturing techniques
3. Classification of the WAAM techniques and their characteristics
4. Overview of WAAM

The other parts (5, 6, 7) are very interesting and well-detailed.
What role does lithium have on the problems of porosity or hot cracking of aluminum alloys?

Regarding the keywords, Al alloys should be replaced by Al-Li alloys.
Also, the main conclusions of this review concerning WAAM processes of aluminum-lithium alloys should be announced in the abstract.

Author Response

Dear Reviewer,

Thank you for your valuable feedback.

We’d like to thank you for taking the time to write the review report of our paper. All feedback is highly appreciated, as it enables us to improve it. All the comments have been made:

We have reduced points 1, 3, and 4, especially point 1, we fully agree with you.

Regarding your question about What role does lithium have on the problems of porosity or hot cracking of aluminum alloys?

You have raised an important point here. The lithium increases the hydrogen solubilization into liquid aluminium. Besides, its high affinity with hydrogen atoms promotes the formation of compounds that would be detrimental in the welding process due to the formation of pores.

In addition to the above comments, all spelling and grammatical errors pointed out by the reviewers have been corrected and we have performed a full English review with a native speaker.

We look forward to hearing from you in due time regarding our submission and to responding to any further questions and comments you may have.

Please see the new version of the manuscript.

Reviewer 2 Report

The paper summarizes the currents state of WAAM deposition on Al alloys with focus on Al-Li alloys. This review sets the WAAM into the context of other DED methods and gives overview of wire for wire manufacturing methods. Then WAAM are systematically characterized and their application summarized in an overview covering the whole history of the process development. Also Al-Li alloys are summarized and the effect of alloying elements described. The welding problems of AL alloys are systematically presented including porosity, hot cracking and humping phenomena and their relevance top Al-Li alloys is discussed. In the last chapter the WAAM techniques already used for the Al-Li alloy deposition are overviewed and results discussed.

The review is on a attractive topic, well written in acceptable English. Only minor points need to be corrected, therefore MINOR REVISION is suggested.

I suggest to try summarizing the mechanical properties of the WAAM alloys is they are available in the references.

Minor comments

L32: “with the addition of wire”, please rephrase “from wire feedstock” etc

L50-52: the classification is by material metal/nonmetal and by parts groups engine, propeller, turbine ? Please rephrase

L70: Chemical composition of wire cannot be modified ….. please  compare with powder, which can be modified ..” in contrast to powder feedstock that can be mixed to……” or similar

L72: “high quality POWDER

L92: “elemental”->”elementary?”

L144: .Concurrently (space)

L144: alloys PRODUCED BY CONFORM…

L180: space before [7]

L224: O.R ?

L244: Siemens controller ?? of substrate positioning?

L385: Besides TI is…..

L433: due to XitsX high AL affinity to oxygen

L439: raises the solubility in SOLID

L434: “give to obtain”

L466: varestraint -> Varestraint

L468: Varestraints

L469: 468-472 needs rephrasing, such as “ maximum crack length in fusion zone as a function of temperature “ or similar

L476-477: please rephrase, you mean that “maximum cracking was observed in solidification range ?“

L479: the alloys CAN BE CHARCTERIZED by plotting

L522: would not alloys above 2.6% of Li be O.K., the crack length is lower in that range ?

L555: CMD abbreviation should be reminded here rather than on L564 below

L574: NO alloy should

L584: space before [104]

L586: fine equiaxed grains don’t imply reduction of porosity

L591: were obtained.

L592: use of less penetration, please rephrase

L597: .).

L603: VP-CMT not defined, abbreviation VP not explained bellow

L627: PMC y CMT

L628: 1mm ?

L659: 8) Conclusions

Author Response

Dear Reviewer,

Thank you for your valuable feedback.

We’d like to thank you for taking the time to write the review report of our paper. All feedback is highly appreciated, as it enables us to improve it.

All the comments have been made.

In addition to the comments, all spelling and grammatical errors pointed out by the reviewers have been corrected and we have performed a full English review with a native speaker.

We look forward to hearing from you in due time regarding our submission and to responding to any further questions and comments you may have.

Please see the new version of the manuscript.

Reviewer 3 Report

This manuscript cannot be further considered before following questions are well addressed.

1.      The language of this manuscript needs to be carefully checked. There are still a lot of works to do for this work.

2.      For a comprehensive review paper, both the experimental and simulation studies should be included in the discussion. But, it seems that this work only focuses on the experiments.

3.      All the inserted figures should be replaced by high-resolution ones. For example, different parts of the Conform machine in Fig.1 cannot be clearly distinguished.

4.      Name of Fig.2 “Figure 2. Classification of the directed process.”

5.      The publications from Prof. Stewart Williams in Cranfield University should be included in a review paper of WAAM of aluminum alloys because this group has made considerable contributions to this field.

6.      Which paper is referenced for Fig.3?

7.      In section 5.2, where is the application of Al-Cu-Li alloy?

8.      More papers on the WAAM of Al-Cu-Li alloys should be included in your review.

Author Response

Dear Reviewer,

Thank you for your valuable feedback.

We’d like to thank you for taking the time to write the review report of our paper. All feedback is highly appreciated, as it enables us to improve it. All the reviews have been considered: 

Comment 1. The language of this manuscript needs to be carefully checked. There are still a lot of works to do for this work.

Response: In addition to the comments, all spelling and grammatical errors pointed out by the reviewers have been corrected and we have performed a full English review with a native speaker.

Comment 2:      For a comprehensive review paper, both the experimental and simulation studies should be included in the discussion. But, it seems that this work only focuses on the experiments.

Response: Thank you for this suggestion. It would have been interesting to explore this aspect. However, in the case of our study, it seems slightly out of scope since a review of simulation studies covers a very wide field. 

Comment 3.      All the inserted figures should be replaced by high-resolution ones. For example, different parts of the Conform machine in Fig.1 cannot be clearly distinguished.

 Response: Agree. We have, accordingly, changed the inserted figures with low quality by new high-resolution images.

Comment 4.      Name of Fig.2 “Figure 2. Classification of the directed process.”

Response: We agree with this and have incorporated your suggestion.

Comment 5.      The publications from Prof. Stewart Williams in Cranfield University should be included in a review paper of WAAM of aluminum alloys because this group has made considerable contributions to this field.Response:

Thank you for pointing this out. We agree with this comment. Thus, we have included the publications from Prof. Stewart Williams in points 4, 6, and 7.

Comment 7.      In section 5.2, where is the application of Al-Cu-Li alloy?

Response: We agree with this, and in section 5.2 we selected the most commonly used alloys in the industry and looked for their applications such as fuselage panels, fuselage frames, and floor beams, among others, we also summarized the alloys mentioned in a scheme (Figure 3).

Comment 8.      More papers on the WAAM of Al-Cu-Li alloys should be included in your review.

Response: Agree. We have included more papers on Al-Cu-Li for WAAM to emphasize this point.

We look forward to hearing from you in due time regarding our submission and to responding to any further questions and comments you may have.

Please see the new version of the manuscript.

Reviewer 4 Report

The manuscript reviews the WAAM of Al-Li alloys. The mauscript has serious flaws and it cannot be considered for publication in the current form:

1- The style and language of the manuscript should be revised.

2- If the subject of the review paper is WAAM, it is not necessary to add the classification of AM techniques especially as a figure. The review shoudl be fully focused on WAAM.

3- The formation of phases in the regular aluminum alloys (not WAAM) should be first explained and discussed and then, you can discuss the changes in a review paper. Please cite the following paper and similar papers and explain the phases and other aspects.

https://doi.org/10.1016/j.corsci.2021.109895

4- General welding defects in aluminum alloy are not related to WAAM. Please delete this section and replace with the defect in WAAM.

5- Fig. 7 is not related to the topic of the manuscript. It should be deleted and replaced by some results about WAAM.

6- The conclusion is not appripriate and could not give an overview about WAAM in aluminum alloys.

Author Response

Dear Reviewer,

Thank you for your valuable feedback.

We’d like to thank you for taking the time to write the review report of our paper. All feedback is highly appreciated, as it enables us to improve it.

All the reviews have been considered: 

Comment 1. The style and language of the manuscript should be revised.

Response: In addition to the comments, all spelling and grammatical errors pointed out by the reviewers have been corrected and we have performed a full English review with a native speaker.

Comment 2. If the subject of the review paper is WAAM, it is not necessary to add the classification of AM techniques especially as a figure. The review shoudl be fully focused on WAAM.

Response:  Thank you for pointing this out. We agree with this comment. We have reduced points 1, 3, and 4, especially point 1.

Comment 3. The formation of phases in the regular aluminum alloys (not WAAM) should be first explained and discussed and then, you can discuss the changes in a review paper. Please cite the following paper and similar papers and explain the phases and other aspects.

Response: Agree. We have, accordingly, described the phases formed with the main alloying elements such as lithium and copper to emphasize this point, and the paper mentioned was cited.

https://doi.org/10.1016/j.corsci.2021.109895

 Comment 4. General welding defects in aluminum alloy are not related to WAAM. Please delete this section and replace with the defect in WAAM.

Response: We agree with this and have incorporated your suggestion. We have included some specific defects in WAAM such as lack of fusion and delamination. We have also modified some terms such as hot-cracking which is widely used for arc welding, by solidification cracking, which is more focused on WAAM defects.

Comment 5. Fig. 7 is not related to the topic of the manuscript. It should be deleted and replaced by some results about WAAM.

Response: You have raised an important point here. However, we believe that it would be more appropriate to mention it because Derekar published a review in 2018 ( https://doi.org/10.1080/02670836.2018.1455012), in which Synchro-feed is cited as a system worth considering in WAAM of aluminium. In the last 5 years, the technology has been available with promising results (See video: Synchro-feed Aluminum wire & arc additive manufacturing (3D printing) Youtube https://www.youtube.com/watch?v=-y40NkKNhtw). Synchrofeed is based on short-circuiting metal transfer similar to CMT and PMC, which are also cited in the review.  

Comment 6. The conclusion is not appripriate and could not give an overview about WAAM in aluminum alloys.

Response: Agree. We have been able to incorporate changes to reflect this suggestion. The conclusions and abstract were modified, and a graphical abstract was also included.

We look forward to hearing from you in due time regarding our submission and to responding to any further questions and comments you may have.

Please see the new version of the manuscript.

Round 2

Reviewer 3 Report

All the comments have been well addressed. It is can be considered for publication now. 

Reviewer 4 Report

The manuscript seems much better than the previous version.